# HPExplainPro: A Framework for Pan-cancer Prognosis Prediction Based on Deep Interpretable Learning

Houwu Gong
College of Computer Science and Electronic Engineering
Hunan University
Changsha Hunan China
hwgong@hnu.edu.cn

Zhijian Huang
College of Computer Science and Electronic Engineering
Hunan University
Changsha Hunan China
zhijian@hnu.edu.cn

Min Jin*
College of Computer Science and Electronic Engineering
Hunan University
Changsha Hunan China
jinmin@hnu.edu.cn

## ABSTRACT

Prognostic prediction research holds immense significance in guiding doctors in evaluating the effectiveness of various treatment modalities, thereby facilitating the selection of the most appropriate treatment plan tailored to individual patients. However, a critical challenge that persists in this domain is the scarcity of clinical interpretability. This issue primarily stems from the inherent opacity of deep learning algorithms, often referred to as "black boxes," where most models operate with closed decision-making processes, lacking transparency in explaining the reasons behind their predictions. To address this gap, this article introduces HPExplainPro, a purpose-built deep explainable learning framework tailored for pan-cancer prognostic prediction. HPExplainPro is composed of a deep learning model rooted in expert knowledge, a data-driven feature fusion approach, a triple feature selection technique, a heterogeneous classifier, and a secondary learning probability error integration model. At its core, HPExplainPro features the Deepxplain module, which leverages global interpretation via DeepSHAP and local interpretation through LIME algorithms to provide insights into the decision-making process. To demonstrate the superiority of HPExplainPro, this article employs three distinct cancer datasets sourced from preeminent hospitals in China. These datasets were leveraged to construct an immunotherapy ORR prediction model for lung cancer, a 5-year survival prediction model for breast cancer patients, and a local progression outcome prediction model for early liver cancer microwave ablation. The experimental results unequivocally demonstrate that HPExplainPro outperforms alternative methods. Furthermore, through Deepxplain's global interpretation capabilities, the study identifies potential prognostic biomarkers such as NETs, LDH, and NLR, which significantly influence the outcome of lung cancer immunotherapy. Additionally, HPExplainPro's local interpretation functionality enables individualized prognostic predictions for lung cancer patients, offering clinicians tailored insights into patient-specific responses to treatment, see in figure 1. Beyond lung cancer, this article explores the broader applicability of HPExplainPro in other diseases. Specifically, it presents a COVID-19 critical illness prediction model using patient data from Wuhan Third Hospital, illustrating the flexibility of HPExplainPro in addressing diverse

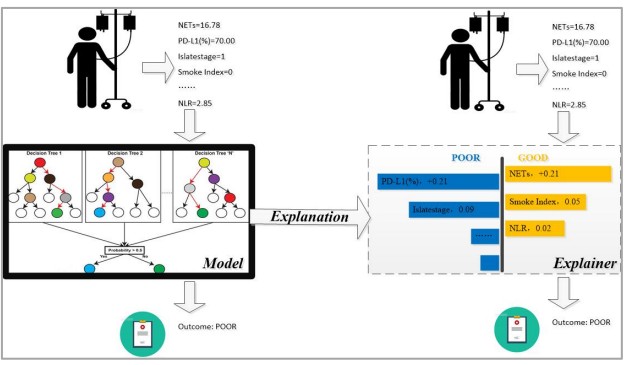

**Figure 1 the local interpretation results of HPExplainPro.**

clinical challenges. Additionally, the study delves into the utilization of HPExplainPro in prognostic prediction within the realm of traditional Chinese medicine (TCM), developing a prognostic prediction model, named "Zhongjing," that harmoniously integrates principles from both TCM and Western medicine. This additional validation further underscores the generalization performance of HPExplainPro across different medical domains. Lastly, the reliability and practicality of HPExplainPro are further bolstered by its validation in the breast department of Hunan Provincial Cancer Hospital. This comprehensive validation process not only validates the effectiveness of HPExplainPro but also showcases its potential to enhance clinical decision-making and improve patient outcomes across a wide range of cancers and diseases.

## CCS CONCEPTS

• Applied computing → Life and medical sciences

## KEYWORDS

Interpretable Learning, Deep Learning, Black Box, Prognosis Prediction, Pan-cancer

## 1 INTRODUCTION

Artificial intelligence (AI) has taken center stage in contemporary medical research, particularly in the diagnosis and treatment of

intricate diseases [1]. The complexities clinicians encounter in crafting treatment plans and forecasting disease outcomes are well-documented [2]. Nonetheless, the swift advancements in AI technology in recent years have revolutionized its application in diagnosing, treating, and, crucially, predicting the prognosis of these complex diseases [3].

When it comes to intelligent prognosis prediction for complex diseases, AI algorithms excel at analyzing vast clinical datasets to uncover disease progression patterns and identify the critical factors that shape patient outcomes [4]. This breadth of algorithms spans machine learning, deep learning, and Bayesian networks. Among these, machine learning algorithms, such as support vector machines, decision trees, and random forests, have demonstrated promising results in predicting the prognosis of diverse diseases [5]. Additionally, deep learning algorithms, including convolutional neural networks and recurrent neural networks, have shown exceptional prowess in predicting prognoses based on image and sequential data, attributed to their robust ability to learn features [6].

**Opportunities.** As cutting-edge technologies in predictive modeling, machine learning algorithms have catapulted medical AI to unprecedented pinnacles, providing vital clinical decision support in domains encompassing disease diagnosis and risk prediction. Nevertheless, despite the remarkable enhancements in predictive model performance stemming from advancements in these algorithms, their intricate and often inscrutable reasoning processes, colloquially termed "black box" thinking, have undermined the trust of end-users [7]. This swift evolution has relegated traditional modeling techniques, such as linear and logistic regression, to the category of legacy algorithms. However, when juxtaposed against the enigmatic decision-making mechanisms of newer algorithms, traditional algorithms that offer explanations for their reasoning processes seem to garner greater trust. Consequently, some researchers argue that AI systems must possess two fundamental interpretability aspects: the ability to elucidate and validate their predictions, as well as the transparency of their knowledge sources to establish trust among physicians [8].

**Challenges.** Current prognostic prediction models continue to encounter significant challenges. Primarily, the predictive accuracy of these models needs significant enhancement to align with the personalized demands of diverse diseases and patients. Secondly, these models are frequently hindered by the "black box" issue, which stems from a lack of sufficient transparency in their decision-making processes, ultimately undermining the trust of doctors and patients in their predictions [9]. Consequently, the development of efficient, accurate, and interpretable prognostic prediction models has become a critical priority in current disease prognosis research [10]. To tackle this challenge, this paper presents a pan-cancer prognostic prediction approach called HPExplainPro, which employs deep interpretable learning for intelligent prognosis forecasting. To assess the efficacy of HPExplainPro, comparative experiments are conducted against various methodologies, particularly focusing on lung cancer,

breast cancer, and liver cancer. Furthermore, this paper explores potential biomarkers that influence pan-cancer prognosis, aiming to provide valuable insights that can contribute to the advancement of related fields.

**Contributions.** In this work, we present HPExplainPro — a comprehensive framework harnessing the power of deep interpretable learning to facilitate accurate pan-cancer prognosis prediction. HPExplainPro has undergone meticulous evaluation, encompassing three cancer-specific experiments, external validations across diverse diseases, and practical clinical applications. Its implementation addresses pivotal scientific challenges inherent in AI methodologies, particularly those pertaining to deep learning. These challenges include limited robustness, scant interpretability, and a heavy reliance on extensive datasets. By successfully overcoming these obstacles, HPExplainPro empowers medical AI to earn greater trust and widespread adoption among users, ultimately enhancing its supportive role in disease treatment and patient rehabilitation endeavors.

## 2 RELATED WORK

In clinical practice, commonly used prognostic prediction methods include Logistic regression and COX regression [11-13]. However, the growing volume of patient data—characterized by its vastness, diversity, and complex structure—poses significant challenges to traditional statistical methods. For instance, Abdulaal et al. [14] sought to develop a model that could provide early mortality warnings upon patient admission. They collected electronic medical records of COVID-19 patients admitted to a local hospital between February and April 2020. Utilizing an Artificial Neural Network (ANN), they conducted an in-depth analysis of various patient characteristics, including demographic information (such as age and gender), comorbidities, smoking history, and a range of symptoms such as the duration since symptom onset, fever, cough, shortness of breath, myalgia, abdominal pain, and diarrhea.The model processes these clinical data inputs to predict the mortality rate at the time of admission. If the predicted mortality probability exceeds 50%, it indicates that the patient's prognosis may be poor. In the test group, the predictive model demonstrated high specificity and sensitivity in assessing patient mortality risk, achieving scores of 0.863 and 0.875, respectively.

Huang Yan from the Breast Surgery Department at the 307 Hospital of the Academy of Military Medical Sciences emphasized that investigating the heterogeneity of breast cancer and its relationship to tumor development can address treatment challenges, predict prognosis, enable personalized treatment assessments, and improve patient survival rates. Although numerous studies have focused on breast cancer prognosis prediction, only the 21-gene recurrence score and the 70-gene signature have gained clinical approval. However, the high costs, technical limitations, and poor reproducibility of these tests

highlight the need for practical and affordable tools for predicting breast cancer prognosis in clinical settings.

Many machine learning models are like mysterious "black boxes," lacking the ability to explain how they arrive at their predictions. This lack of transparency can pose challenges for doctors, especially when the model's suggestions clash with their own intuition, leaving them puzzled without any rationale. Enhancing the clarity of these models is crucial for boosting their reliability and effectiveness in aiding medical decision-making. Take predicting ICU mortality rates, for example. The model's diagnostic findings play a significant role in shaping patient care. A model that can be easily understood can pinpoint the key factors affecting mortality, thereby offering stronger support for clinical judgment. This empowers doctors to more accurately assess the decision's validity and enables them to intervene in the patient's treatment process earlier.

The lack of clear clinical interpretability poses a significant challenge when it comes to predicting disease prognosis. In response, this paper introduces HPExplainPro, a holistic method for predicting cancer prognosis. HPExplainPro combines transformer architecture with interpretability techniques, boosting model effectiveness in several key areas: data preparation, feature engineering, prediction model, decision model, and interpretation of results. This strategy offers a comprehensive insight into the deep learning model from both a broad overview and specific details.

# 3 THE HPEXPLAINPRO FRAMEWORK

Figure 2 illustrates the complete workflow of the HPExplainPro model outlined in this paper. It is broadly divided into five main stages.

**Stage 1:** To prepare the data: we gather basic information, medical history, diagnostic records, examination results, electronic medical records, and patient follow-up data from hospital information systems. Next, we split the clinical data into training, validation, and test sets using a 10-fold cross-validation technique, ensuring that the test set remains unchanged for feature selection and model training. Additionally, we conduct data preprocessing simultaneously to prepare the data for the feature selection stage.

**Stage 2:** Feature Engineering: In the initial phase, we use techniques like feature transformation and feature combination to unveil underlying feature patterns. In the subsequent phase, we apply three feature selection algorithms—chi-square test, t-test combined with mutual information, maximum relevance minimum redundancy algorithm, and genetic algorithm—to refine the data features, resulting in three optimal feature subsets. After feature processing, we evaluate the balance of the data samples. If any imbalance is detected, corrective measures are implemented to address it.

**Stage 3:** Prediction Model: we divide the reduced learning set into multiple training sets using N-fold cross-validation. These diverse

training sets are used to train four different algorithms, which include deep models such as CNN and transformer. Through this process, we obtain a multi-heterogeneous classification model and determine class weights for each model category.

**Stage 4:** Decision Model: The diverse classifiers produce class probability predictions for each fold's validation set, leveraging the optimal parameters derived in the preceding step. These predictions, along with the actual class labels, are amalgamated to create a new dataset. Following this, we employ XGBoost for secondary learning on this dataset to fine-tune the probability errors. Ultimately, we establish an ensemble model comprising heterogeneous classifiers. Subsequently, diagnostic predictions for unknown samples are made through competition based on class weights.

**Stage 5**: Prediction Results: The Deepxplain model is employed to provide explanations for the model from both a global and local standpoint. To obtain the final classification diagnosis results, each fold of the test set is subjected to testing, employing the feature subset selected through triple feature selection for every fold of the test set.

## 3.1 Data Preparation

In disease prognosis prediction, data preparation is fundamental. It involves gathering essential patient information such as age, gender, and medical history, which serves as the foundation for building predictive models. Furthermore, diagnostic and treatment data, including treatment plans and medication records, are essential as they document the patient's treatment journey. Additionally, diagnostic test data such as imaging and biochemical indicators act as vital reference points for prognosis prediction. Follow-up data documenting changes in the patient's condition after discharge are also critical for assessing predictive model accuracy. By integrating these diverse datasets, a comprehensive understanding of the patient's condition can be attained, providing robust data support for prognosis prediction. Building upon this, the paper develops a comprehensive, multidimensional clinical dataset by seamlessly integrating these data, thus enhancing the understanding of the complex synergistic effects among risk factors in the prognosis process of complex diseases.

**Data preprocessing methods.** Upon acquiring clinical data, preprocessing becomes imperative due to the frequent presence of missing or outlier values in the raw data, which can detrimentally affect the predictive capability of models. Key preprocessing techniques encompass addressing outliers and handling missing values. Outliers often manifest as special symbols (e.g., "<", "-") or erroneous data (e.g., negative age values). Managing outliers involves either replacing them or eliminating them entirely. Dealing with missing values encompasses either removing features with missing values or imputing them. Specifically, features with an excessive number of missing values are removed, and if a single indicator contains over 50% missing values, the

entire sample information is excluded. Missing samples are then filled by using the median or mean for continuous features, mode for categorical features, or through methodologies such as the k-

frequently arise in clinical data for various reasons. For instance, certain phenomena may appear or not appear in different environments, contributing to these imbalances. To enhance the performance of machine learning classifiers and improve classification accuracy and recognition capabilities, it is crucial to address the estimation issues caused by imbalanced data sets. The primary methods for handling imbalanced data sets involve sampling techniques at the data level, which include both upsampling and downsampling. At the algorithm level, optimization involves considering the cost differences associated with various misclassification scenarios. The following section provides a detailed overview of the techniques used to manage imbalanced data sets in this paper, with a focus on data-level approaches.

Adhering to the "randomized, controlled, double-blind" principles in the medical field, the concept of sampling emphasizes "randomness and balance." When there is a significant disparity in the sample sizes of different classes within a dataset, trained models often become biased towards the more prevalent class. To improve the learning from minority class samples, it is

nearest neighbors (KNN) algorithm.

**Methods for Handling Imbalanced Data.** Imbalanced data sets

essential to address the issue of imbalanced data. The primary methods for this include: Synthetic Minority Over-Sampling Technique (SMOTE) and Random Under-Sampling (RUS). Upsampling involves increasing the number of minority class samples to achieve a more balanced dataset. This method uses interpolation on existing minority class sample data to create new samples, which are then added to the dataset. For each sample in the minority class, the K-Nearest Neighbors (KNN) algorithm calculates the distances to other samples in the same class, identifying the k-nearest neighbors. Based on the imbalance ratio, a sampling rate is determined, and neighboring samples are randomly selected. These selected neighbor samples, together with the original sample, are used to generate new sample data through interpolation. Downsampling involves reducing the number of majority class samples to achieve a more balanced dataset. The downsampling method, Random Under-Sampling (RUS), randomly selects a subset of the majority class samples and removes them from the dataset.

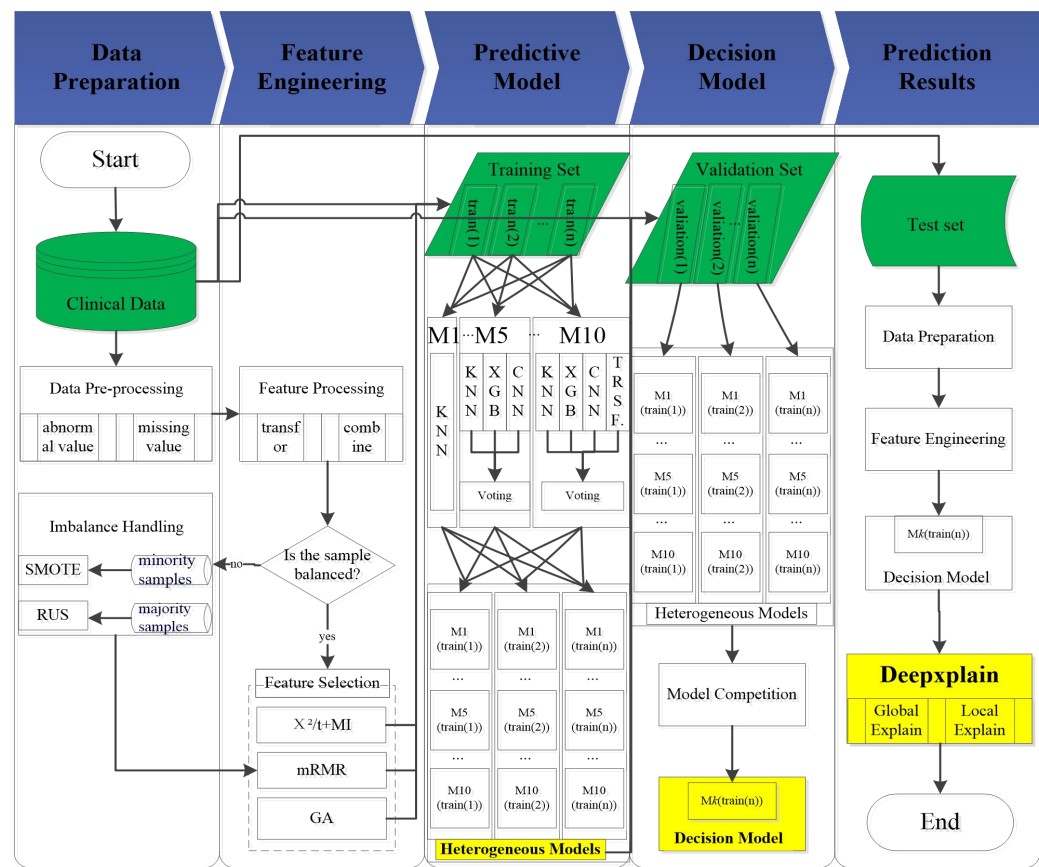

**Figure 2: HPExplainPro Model Framework.**

## 3.2 Feature Engineering

The HPExplainPro model's feature engineering module integrates expert knowledge-based and data-driven feature fusion techniques, enhanced by a triple feature selection strategy.

**An approach to feature fusion that combines expert knowledge with data-driven techniques.** Feature transformation, combination, and other processing techniques enable the exploration of underlying information hidden within features. Feature fusion not only consolidates data from diverse dimensions but also uncovers intrinsic relationships and potential patterns among features. This approach not only enhances diagnostic accuracy but also aids in the early detection of lung cancer. For instance, utilizing basic measurements like height and weight to calculate BMI offers insights into a patient's overall health. Nevertheless, for more precise lung cancer diagnosis, expertise is paramount. Medical professionals advocate for the neutrophil-to-lymphocyte ratio (NLR) as a pivotal blood biomarker, indicating potential inflammatory responses or immune status within the patient's body. Therefore, employing a data-driven approach to meticulously analyze this ratio substantiates its importance in lung cancer diagnosis.

**Method for triple feature selection.** The clinical dataset for early diagnosis of complex diseases presents a multimodal nature, characterized by high dimensionality and significant variations across dimensions. In actual clinical datasets, only a few features exhibit a strong correlation with early disease diagnosis, amidst numerous noisy and redundant features. Consequently, addressing cancer classification in such multimodal clinical datasets poses a critical and formidable challenge. Filter methods, which can be applied across various machine learning algorithms and are not tied to specific learners, offer remarkable versatility. They demonstrate computational efficiency, making them suitable for large-scale datasets and scenarios requiring rapid results. Conversely, wrapper methods involve substantial computational overhead and complexity, potentially making them impractical for large-scale datasets.

Traditional filter methods are commonly employed due to their high generality and low complexity, which facilitate the rapid reduction of data dimensionality. However, their limitation arises from the assumption of linear separability within the data. Clinical data, known for its high redundancy, may not always adhere to this assumption. To address this challenge, our paper proposes three feature selection methods: the chi-square test, t-test combined with mutual information, maximum relevance minimum redundancy algorithm, and genetic algorithm. These methods are designed to optimize data features, resulting in three optimal feature subsets. More detailed explanations are provided below:

The primary feature selection algorithm: Using a statistical significance-based approach to feature selection, we combine the chi-square test, t-test, and mutual information technique to reduce the dimensionality of the clinical dataset and remove irrelevant features. The chi-square test compares categorical variables, the t-test compares continuous variables, and the mutual information method assesses the correlation between variables. By integrating these methods, we can effectively pinpoint features with substantial differences, thus improving the precision and efficiency of disease diagnosis.

The second-tier feature selection algorithm: We've opted for the mRMR algorithm due to its efficiency in eliminating redundant features while maximizing correlations within clinical data. At its core, this algorithm aims to enhance the relevance between features and classification targets to identify the optimal m features. However, the mere presence of these m features does not guarantee the highest predictive accuracy. Let's define the data matrix $D = \{x_{i,j}, y_i (x_{i,j} \in R^n, y_i \in y)\}_{M \times N}$, where $x_{i,j}$ signifies the j-th feature of the i-th sample, with $R^n$ representing the feature space, and y denoting the classification label set. We denote the selected feature set as S. By employing specific formulas, the mRMR algorithm enhances feature-classification correlations while reducing inter-feature correlations:

$$max \left( \frac{1}{|S|} \sum_{x_i \in S} I(x_i; t) - \frac{1}{|S|^2} \sum_{x_i, x_j \in S} I(x_i; x_j) \right) \qquad (1)$$

The third-tier feature selection algorithm: Using a genetic algorithm, this approach imitates optimization through natural selection and genetic mechanisms, possessing robust global search capabilities, which render it suitable for addressing various complex optimization problems. By iteratively refining the process, this paper automatically identifies the most representative features, thus enhancing the accuracy and efficiency of the diagnostic model.

## 3.3 Heterogeneous Classifier

The HPExplainPro model brings together two distinct deep learning architectures, CNN and Transformer, alongside traditional machine learning techniques and ensemble methods, to create a diverse classifier. CNN excels in tasks involving image processing and computer vision, efficiently extracting local image features through convolutional operations and gradually building higher-level abstract representations with hierarchical structures. This prowess has led to significant progress in image classification and object detection. However, CNN encounters difficulties with sequential data due to its focus on local features, hindering its ability to capture global dependencies. In contrast, Transformer has achieved notable success in natural language processing endeavors. By leveraging self-attention mechanisms, it effectively captures long-range dependencies within sequences, facilitating the adept handling of contextual language information. This achievement is particularly evident in tasks such as machine translation and text generation. Nonetheless, Transformer's computational complexity presents a challenge, especially when processing lengthy sequences, resulting in heightened resource consumption. When it comes to disease prognosis prediction, harnessing the strengths of both CNN and Transformer proves advantageous. CNN excels in extracting local features from patient image data, such as medical images, capturing the localized manifestations of diseases. Conversely, Transformer

adeptly processes sequential data like patient medical records and examination results, capturing the global dependencies essential for predicting disease prognosis. The synergistic integration of these models effectively leverages their respective strengths, thereby enhancing the accuracy and reliability of disease prognosis prediction.

In summary, the heterogeneous classifier overcomes the constraints of individual models and optimizes their benefits by combining the complementary features of CNN and Transformer, two distinct deep learning models. This strategy effectively harnesses the strengths of both models, offering strong support for disease prognosis prediction.

## 3.4 Integrated Model of Secondary Learning Probability Error

The prevailing ensemble techniques encounter challenges in capturing the nonlinear relationships among different classifiers and struggle to effectively grasp and precisely estimate the nonlinear variations in clinical data after dimensionality reduction. Addressing this issue, this paper introduces an innovative ensemble strategy centered on second-order learning of probability errors.

This strategy begins by amalgamating the category probability forecasts of multiple heterogeneous classifiers on the validation set with the true category labels post the initial learning phase, thereby crafting a comprehensive new dataset. This method ingeniously converts the classification errors produced by individual models into probability format, amalgamating them into the newly formed dataset. Subsequently, we proceed with a second round of learning using this dataset.

In this subsequent learning phase, we opted for the XGBoost algorithm as the cornerstone tool, leveraging its expansion of the Gradient Boosting algorithm's capabilities and its popularity owing to its remarkable error learning and nonlinear fitting prowess. The core concept behind XGBoost lies in using residuals between predicted and true values as the learning objective for subsequent iterations, progressively minimizing these residuals to enhance model performance. Additionally, the algorithm incorporates L1 and L2 regularization terms into the objective function of each iteration, effectively managing model complexity and mitigating overfitting.

Considering the potentially substantial sample size of individual disease samples in clinical datasets, we selected the XGBoost algorithm, which does not hinge on large sample sizes, to ensure the effectiveness and efficiency of second-order learning. By thoroughly learning and fitting the errors between the predicted probabilities of individual models from heterogeneous classifiers in the new dataset and the true values, we successfully attained an unbiased estimation of the nonlinear integration of heterogeneous classifiers and the nonlinear data change patterns.

In summary, this ensemble strategy, based on second-order learning of probability errors, actively forecasts, learns, and rectifies the classification errors of multiple heterogeneous classifiers, providing robust support for enhancing the predictive performance of disease diagnosis. Through the integration of heterogeneous classifiers, the ensemble model proposed in this paper is realized.

## 3.5 Deepxplain Interpretable Model

The DeepXplain explanation model empowers HPExplainPro with a comprehensive and detailed tool for understanding the predictive behavior of machine learning models through two methods: global explanation using DeepSHAP and local explanation using LIME. Global explanation aims to delve into the overall decision logic and functioning of the model, providing users with a thorough and detailed explanation of the sample on a global scale. It scrutinizes the influencing factors in the feature subset and identifies potential biomarkers affecting the target outcome. Conversely, local explanation focuses on investigating the model's decision-making process for individual instances, providing specific decision logic or basis of the machine learning model for each input sample to clarify the reasons behind the classification results.

**Global interpretation.** Shapley Additive Explanations (SHAP) is a technique that elucidates the outcomes of machine learning models by integrating expected values with Shapley values [15]. It evaluates the relationship between each feature element and the model's predictive capacity, along with its impact on the prediction results, thereby assisting individuals in understanding the practical significance encapsulated by model features. The SHAP value assigned to each feature serves as the cornerstone for interpretation, indicating the feature's contribution to the prediction risk of complications. Positive SHAP values indicate that the corresponding feature contributes to an increase in the risk of complications, while negative SHAP values suggest a decrease in the risk of complications associated with the feature.

This section utilizes an improved SHAP explanation tool to provide an extensive explanation of the target prediction model. The enhanced SHAP explanation tool calculates the attributes of each input-baseline pair using a baseline distribution and then averages the resulting attributes for each input. Additionally, it treats the network output as a linear combination of simplified inputs:

$$F(f_X(X')) = \phi_0 + \sum_{n=1}^{N} \phi_n x'_n \qquad (1)$$

In this scenario, X represents the feature vector of the input model, where X belongs to $\{x_1, x_2, ..., x_n\}$, and X' stands for the simplified feature vector of X, with X' belonging to $\{0, 1\}$, indicating whether corresponding features are present in X or not. The variable n denotes the feature dimension, while $\varnothing_n$ serves as a constant. $\varnothing$ represents the feature weights, which can take on positive or negative values, providing a measurement of the respective feature relationships. Additionally, $\varnothing_n$ represents the feature weight of feature vector $x_n$. N represents the total number of features. The scalar output of the model is denoted by $F(f_x(X'))$, and $(f_x(X'))$ denotes the mapping function responsible for converting the binary vector back to the original space.

**Local explanation.** In this section, we provide an overview of the established predictive model and create a practical visualization model for interpreting patient-level data, facilitating comparisons between a selected patient and those without specified complications. The Local Interpretable Model-Agnostic Explanations (LIME), introduced by Marco Ribeiro et al., acts as a valuable tool for comprehending the decision-making process of intricate black-box models [16]. Let's denote the complex model under scrutiny as f and the simpler model as g. Thus, the objective function employed to assess the contrast between these two models can be expressed as:

$$\xi(x) = \sum_{z,z'} \pi_x(z)(f(z) - g(z'))^2 + \Omega(g) \quad (2)$$

The equation depicts $\pi_x(z)$ as the measure of similarity between samples before and after perturbation. In this context, x signifies the sample necessitating elucidation, x' is the sample obtained by removing features to transform into x', z is the fresh sample point generated post perturbation, and z' incorporating excluded features is reverted to the initial sample z. $\Omega$ denotes the complexity of the model.

**Table 2: Prognostic datasets for three different types of cancer.**

| Dataset | Poor prognosis | Good prognosis | Total number | Features |
|---|---|---|---|---|
| Lung cancer | 90 | 114 | 210 | 26 |
| Breast cancer | 60 | 250 | 310 | 33 |
| Liver cancer | 90 | 229 | 319 | 23 |

The second equation highlights that the perturbed samples generated by LIME may lack sufficient validity, potentially impeding the surrogate model's ability to accurately mirror the intricate predictive patterns of the complex model surrounding the focal instance. This research endeavors to refine the methodology for selecting perturbed samples. By augmenting the likelihood of selecting perturbed samples exhibiting higher degrees of validity, we indirectly penalize those with lower validity, thereby refining the precision of LIME's explanatory outcomes. To achieve this objective, we propose a sampling technique that amplifies the chances of selecting samples with superior validity while diminishing the probability of choosing those with inferior validity, thus ensuring the coherence of the perturbed samples selected. The workflow of the LIME algorithm is illustrated in Table 1.

**Table 1: The workflow of the LIME algorithm.**

| Algorithm 1: LIME Algorithm |
|---|
| *Input: 1) the Complex model, f; 2) the Sample of interest, x; 3) the Number of randomly generated samples, N.* |
| *Output: Weights of linear models* |
| *1. Through feature screening, obtain d' important features and obtain the explanatory version of the sample of interest, x';* |
| *2. Perform random perturbation near sample x' to generate new sample data z', restore z' 'to a sample z with the same dimension as x, and use the complex model to predict labels;* |
| *3. Fit the newly generated dataset using a linear model.* |

# 4 EXPERIMENTS

## 4.1 Datasets

This study utilizes datasets from three prominent large hospitals in China, spanning three distinct cancer types: lung cancer, breast cancer, and liver cancer. The lung cancer dataset, obtained from Hunan Cancer Hospital, includes demographic characteristics, symptoms, and blood biomarkers. These encompass age, height, weight, smoking history, pathological subtype, blood biomarkers such as neutrophils, lymphocytes, CD8+ T lymphocytes, absolute and percentage values of CD8+ T lymphocytes, late-stage indicators, liver metastasis, bone metastasis, results of neutrophil extracellular traps (NETs) secondary detection, and objective response rate (ORR). The breast cancer dataset, sourced from Hunan Cancer Hospital, comprises patient age, pathological subtype, maximum tumor diameter, lymph node metastasis, vascular cancer embolism, whether it is classified as triple-negative breast cancer, treatment recommendations, and 5-year survival status. The liver cancer dataset, obtained from the General Hospital of the People's Liberation Army, includes patient demographic details, preoperative laboratory assessments, and ablation parameters. Specific features include basic demographic information, ablation time, power (W), number of ablations, ablation sites, surgical duration, postoperative hospital stay duration, complications, local recurrence, and time of local recurrence. Data summaries of the three different cancer datasets are provided in Table 2.

Feature selection was conducted through a rigorous feature engineering process, integrating clinical expertise and insights from collaborating specialists. The final set of features was carefully determined. Initially, the lung cancer dataset included 26 features, which were narrowed down to the top 13 through feature selection. Based on clinical expert feedback, further refinements were made: neutrophils and lymphocytes were combined into the NLR, and for the CD8 marker, the percentage value was preferred over the absolute value. Additionally, for NETs, the first test result was used if secondary detection values were present. Consequently, the top 10 selected features were: CD8_%, NK, LDH, metastasis sites, NETs, bone metastasis, pathological types, IL-8, NLR, and smoking index.

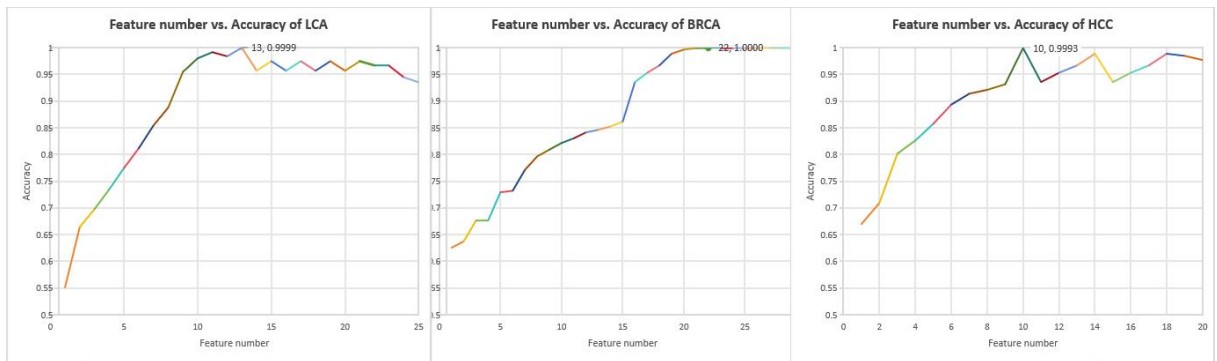

**Figure 3: depicts the correlation curve between the number of features and accuracy across three different types of cancer.**

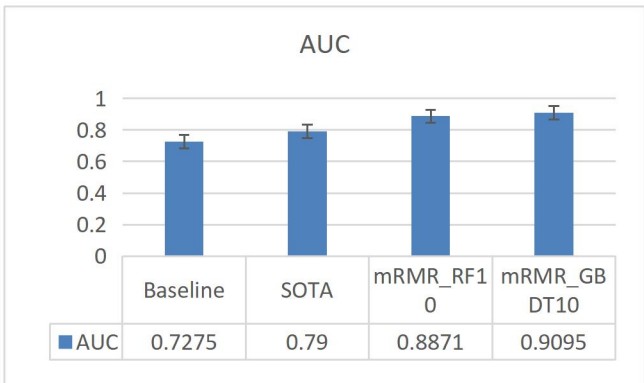

**Figure 4: AUC Comparison  HPExplainPro VS SOTA.**

## 4.2 Evaluation Metrics

To ensure robust test results, this study employs a 10-fold cross-validation method. The clinical dataset is initially divided randomly into 10 parts. In each iteration, 9 parts are used as the training set, while the remaining part serves as the test set. Additionally, within each fold of the 10-fold cross-validation, a K-fold cross-validation is conducted. In this step, the training set is further divided into K mutually exclusive subsets. For each iteration of the K-fold cross-validation, K-1 subsets are used for training, and the remaining subset is used for validation.

**Table 3: Number of samples after preprocessing in the three different cancer datasets.**

|  | Prediction: Normal | Prediction: Tumor |
|---|---|---|
| Actual: Normal | TN | FP |
| Actual: Tumor | FN | TP |

In this study, we consider primary tumor tissue as positive and normal solid tissue as negative. The evaluation of the classifier relies on a confusion matrix, as shown in Table 3, which illustrates the correspondence between actual and predicted classes. The cancer diagnosis results from the test set present four scenarios: true positive (TP), true negative (TN), false negative (FN), and false positive (FP). TP represents the correctly classified samples of tumor tissue, TN denotes the correctly classified samples of normal tissue, FN indicates samples predicted as normal tissue but are actually tumor tissue, and FP represents samples predicted as tumor tissue but are, in reality, normal tissue. The evaluation metrics, including accuracy, sensitivity, specificity, and F1-score, are defined as follows:

$$Accuracy = \frac{TP+TN}{TP+FP+TN+FN} \tag{3}$$

$$Sensitivity = \frac{TP}{TP+FN} \tag{4}$$

$$Specificity = \frac{TN}{TN+FP} \tag{5}$$

$$F1-score = \frac{2TP}{2TP+FP+FN} \tag{6}$$

## 4.3 Hyperparameters Configuration

During the feature selection phase, we applied the mRMR method. We defined the feature variable parameter, denoted by λ, with a step size set to 1. Starting from 1, we iteratively determined the optimal number of features while plotting the relationship curve between feature variables and cancer prognosis. When training diverse classifiers, we automatically optimized the parameters of each classifier through grid search. In this study, the Transformer's num_layers parameter ranged from {6,12,24}, the d_model parameter ranged from {512,1024}, and the learning_rate ranged from {0.001,0.0001}.

To evaluate the broad applicability of our proposed method, we conducted comprehensive investigations on lung cancer, breast cancer, and liver cancer. Based on the specified parameter configurations, we meticulously recorded the accuracy corresponding to varying numbers of features, denoted as n. Figure 3 vividly illustrates the relationship between feature count and accuracy across these three cancer types. As the feature count,

**Table 4: Prognostic Prediction Results (%) for Three Types of Cancer.**

| Evaluation Metrics | Lung Cancer | Breast Cancer | Liver Cancer |
|---|---|---|---|
| No.of features | 13 | 22 | 10 |
| $\begin{bmatrix} TN & FP \\ FN & TP \end{bmatrix}$ | $\begin{bmatrix} 90 & 0 \\ 0 & 114 \end{bmatrix}$ | $\begin{bmatrix} 60 & 0 \\ 0 & 250 \end{bmatrix}$ | $\begin{bmatrix} 90 & 0 \\ 1 & 228 \end{bmatrix}$ |
| Accuracy | 99.99% | 100% | 99.93% |
| Sensitivity | 99.99% | 100% | 99.97% |
| Specificity | 100% | 100% | 100% |
| F1-score | 99.99% | 100% | 99.95% |

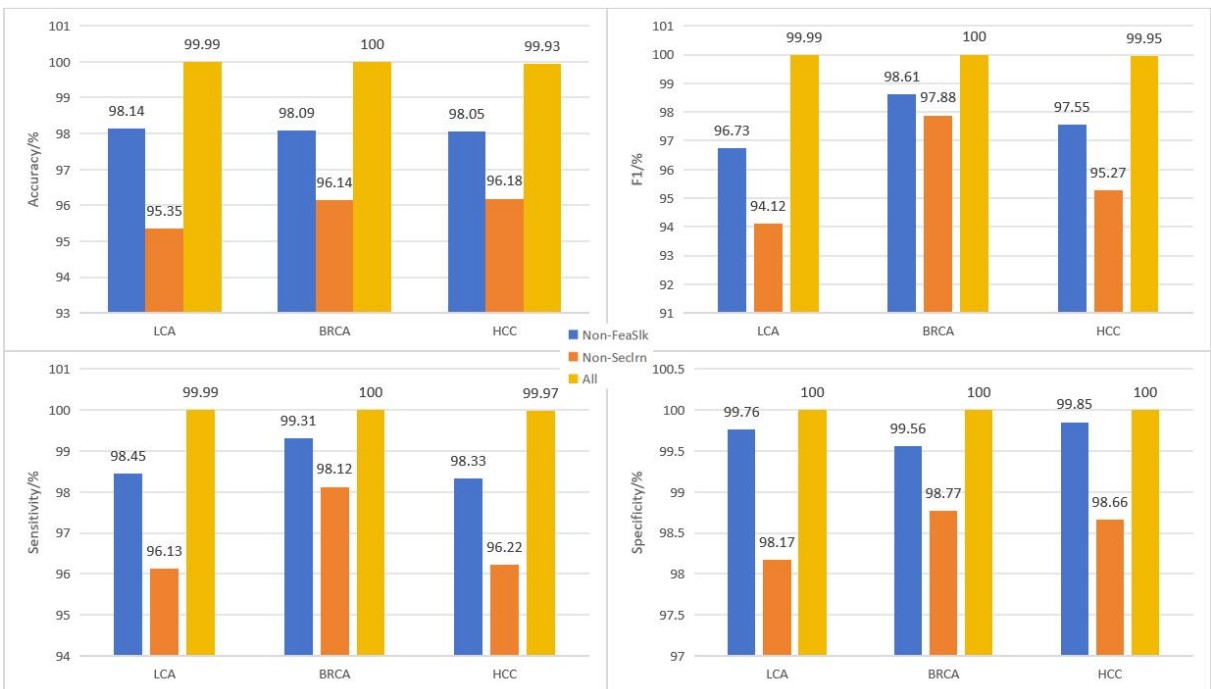

**Figure 5 presents the comparative results of the HPExplainPro ablation experiments.**

n, increases, accuracy initially rises until reaching peak values at feature counts of 13, 22, and 10 for lung cancer, breast cancer, and liver cancer, respectively, followed by a gradual decline. This pattern suggests that beyond a certain number of features, the noise introduced by additional features outweighs their utility. Consequently, in our study, we selected the number of features that initially achieved peak accuracy—13, 22, and 10 for lung cancer, breast cancer, and liver cancer, respectively—as the final feature selection counts.

## 4.4 Main Results

The HPExplainPro model framework has developed predictive models for assessing the objective response rate (ORR) post-immunotherapy in lung cancer, predicting 5-year survival rates among breast cancer patients, and forecasting local progression outcomes following microwave ablation in early-stage liver cancer. Examination of Table 4 yields the following insights: In lung cancer, the utilization of 13 core features resulted in

achieving remarkable metrics, including 99.99% accuracy, 99.99% sensitivity, perfect 100% specificity, and a 99.99% F1 score. For breast cancer, employing 22 key features led to achieving perfect scores across all four metrics. Similarly, in liver cancer, leveraging 10 key features yielded exceptional results with 99.93% accuracy, 99.97% sensitivity, 100% specificity, and a 99.95% F1 score. These experimental findings underscore the high accuracy of HPExplainPro in prognostic predictions across these prevalent cancers using real clinical data. Moreover, this methodology demonstrates extensive applicability and robustness in predicting outcomes across diverse cancer types.

Furthermore, this paper compares HPExplainPro's performance with that of cutting-edge methods (see Figure 4). Chowell et al [17]. introduced RF16, an ensemble learning-random forest classifier with 16 input features, designed to predict the prognosis of various cancers. Their work was published in Nat Biotechnol. The study highlighted RF16's performance: on the training set - (pan-cancer AUC: RF16 0.85, RF11 0.79, TMB

0.62), and on the test set - (pan-cancer AUC: RF16 0.79, RF11 0.71, TMB 0.63).

The study findings reveal a significant improvement in AUC values compared to both the baseline and state-of-the-art (SOTA) methods discussed in this paper. Employing the same Random Forest base model, the method proposed in this section, mRMR_RF10, demonstrated a 15.96% enhancement over the baseline and a 9.71% improvement over SOTA. Additionally, the GBDT-based approach, mRMR_GBDT10, showcased superior performance as outlined in this study. These results underscore HPExplainPro as a robust framework for prognosticating pan-cancer outcomes.

## 4.5 Ablation Study

The methodology outlined in this paper primarily consists of two key components: a triple feature selection algorithm and a secondary learning probability error ensemble model. This section delves into the specific contributions of these components within HPExplainPro by conducting comparisons between the method with and without them. The analysis demonstrates that HPExplainPro achieves optimal performance when both the triple feature selection algorithm and the secondary learning probability error ensemble model are utilized (refer to Figure 6). For instance, in the case of lung cancer, excluding the triple feature selection algorithm results in improvements of 1.85% in accuracy, 1.54% in sensitivity, 0.24% in specificity, and 3.26% in F1-score with HPExplainPro. Similarly, when the secondary learning probability error ensemble model is omitted, HPExplainPro exhibits enhancements of 4.64% in accuracy, 3.86% in sensitivity, 1.83% in specificity, and 5.87% in F1-score for lung cancer. These experimental findings underscore HPExplainPro's efficacy in improving the efficiency of cancer prognosis prediction.

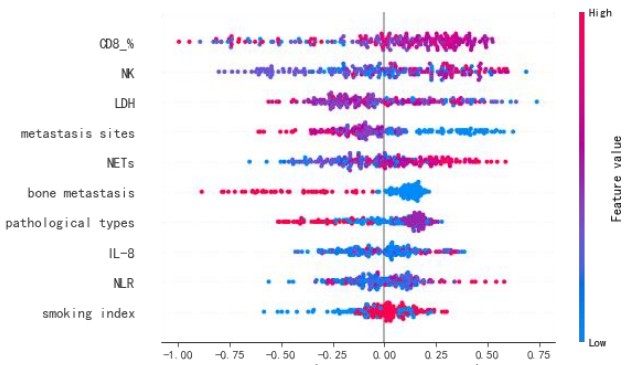

**Figure 6: Global Explanation Results of HPExplainPro.**

## 4.6 Potential Biomarkers for Lung Cancer Prognosis Discovered by DeepXplain

Using the HPExplainPro method, this study sought to forecast the efficacy of immunotherapy ORR in non-small cell lung cancer (NSCLC). The methodology entailed amalgamating clinical insights and integrating variables such as the neutrophil-to-lymphocyte ratio (NLR), CD8 percentage value, and the initial test value of NETs. Through Deepxplain global interpretation, potential biomarkers impacting disease prognosis were pinpointed, encompassing NETs, LDH, CD8, IL-8, metastasis sites, and NLR. Clinical validation was conducted at the hospital [18]. See in Figure 6.

Deepxplain's local explanation illuminates how the HPExplainPro model predicts the prognosis for individual NSCLC patients. Since clinicians may struggle to understand how the model determines a poor prognosis based on patient demographics, clinical features, and test results, Figure 7 elucidates the decision criteria of the black-box model. It emphasizes that a higher score is linked to the programmed death-ligand 1 value (%) and whether the patient is in an advanced stage, ultimately leading to a prediction of a poor prognosis.

## 4.7 External Validation: COVID-19 Critical Illness Prediction

This section details the validation process of HPExplainPro for other diseases. Severe cases of COVID-19 pose a considerable risk of mortality, with 6-8% of confirmed patients advancing to critical illness requiring intubation or ICU care, and the mortality rate for these critically ill patients can soar as high as 65%. Drawing upon data from 1590 COVID-19 cases nationwide, a team led by Academician Zhong Nanshan from the Guangzhou Institute of Respiratory Health integrated clinical risk factors to devise the COVID-GRAM critical illness prediction model for COVID-19 [19]. This model encompasses 10 key features, encompassing abnormal chest X-ray findings, age, hemoptysis, dyspnea, altered consciousness, number of comorbidities, history of previous cancer, neutrophil-to-lymphocyte ratio, lactate dehydrogenase levels, and direct bilirubin levels. COVID-GRAM exhibits the ability to predict whether COVID-19 patients will progress to critical illness with an accuracy rate of up to 88%.

### 4.7.1 Dataset

This section details the development of a critical illness prediction model for COVID-19, utilizing the HPExplainPro framework and data collected from 246 COVID-19 patients at the Third Hospital of Wuhan. The model encompasses various demographic factors like gender, age, weight, and height, as well as medical histories categorized by systems including respiratory, digestive, endocrine, and metabolic diseases, as well as musculoskeletal and reproductive system conditions. Furthermore, it incorporates surgical and infectious disease histories, alongside admission-recorded vital signs such as respiratory rate, body temperature, pulse, and blood pressure. Additionally, the model incorporates laboratory test results like routine blood cell counts, chemical profiles, coagulation profiles, electrolytes, and brain natriuretic peptide (BNP) levels as predictive features.

### 4.7.2 Main Results

In the development cohort, COVID-GRAM achieved an AUC of 0.88, and it also reached an AUC of 0.88 in the validation cohort. HPExplainPro obtained an AUC of 0.92 in the development cohort and 0.90 in the validation cohort. As depicted in Figure 7,

the experimental findings indicate that HPExplainPro outperforms COVID-GRAM in predictive performance.

### 4.7.3 Potential biomarkers for COVID-19 critical illness discovered by DeepXplain

Deepxplain's analysis revealed several factors influencing the severity of COVID-19 critical illness, including age, medical history, neutrophil-to-lymphocyte ratio (NLR), and BNP. Research has consistently shown a strong correlation between age and the severity of COVID-19 symptoms. With age, the immune system gradually weakens, making older individuals more susceptible to viral infections and often experiencing more severe illness, particularly among those aged 70 and above. Additionally, patients' medical history, especially the presence of chronic conditions like cardiovascular disease, diabetes, and hypertension, significantly impacts the severity of COVID-19 symptoms. These chronic conditions can compromise the immune system, heightening the risk of viral infection and exacerbating the illness. Moreover, certain chronic conditions may impair patients' cardiopulmonary function, making them more vulnerable to respiratory attacks from COVID-19. The neutrophil-to-lymphocyte ratio (NLR) is a crucial indicator reflecting the body's inflammatory response and immune status. Studies have demonstrated a close relationship between elevated NLR and the severity of COVID-19 symptoms. During viral infections, neutrophil levels surge to combat the virus, while lymphocyte levels may decrease due to viral attacks. Consequently, an elevated NLR may signify a severe inflammatory response and immune suppression, aligning with the condition observed in critically ill COVID-19 patients. BNP, a hormone produced by the heart, primarily regulates fluid balance and blood pressure. In cardiovascular diseases such as heart failure and myocardial infarction, BNP levels typically rise significantly. Similarly, in critically ill COVID-19 patients, heart muscle damage and cardiac dysfunction resulting from viral attacks can lead to elevated BNP levels. Therefore, BNP serves as a vital indicator for evaluating the cardiac function of COVID-19 patients and predicting the severity of the illness.

In conclusion, age, medical history, NLR, and BNP are potential biomarkers influencing the severity of COVID-19 critical illness. By monitoring changes in these indicators, physicians can more accurately assess the severity of patients' symptoms, devise more precise treatment strategies, and ultimately enhance treatment outcomes and reduce mortality rates.

## 4.8 External Validation: Zhongjing

This section delineates how HPExplainPro is utilized to forecast the prognosis of diseases under traditional Chinese medicine (TCM) treatment. It introduces a prognostic model called "Zhongjing," which harmonizes both Chinese and Western medicine approaches. The objective is to enhance the validation of the model's overall performance.

### 4.8.1 Dataset

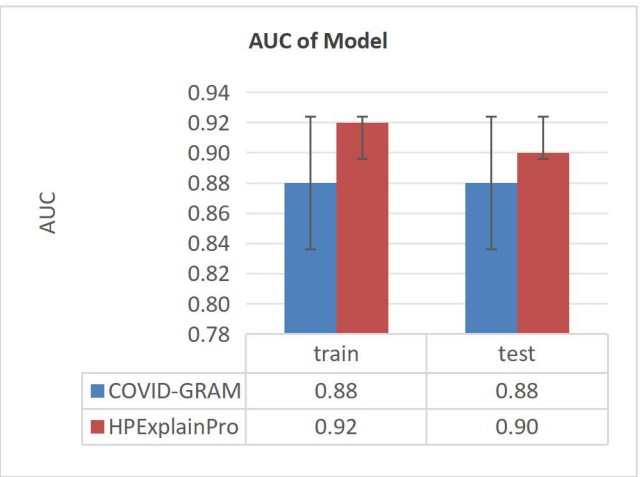

**Figure 7 illustrates the cross-validation results of HPExplainPro.**

The research collected electronic medical records data from 317 osteoarthritis patients treated at the First Affiliated Hospital of Hunan University of Chinese Medicine. All participants underwent significant orthopedic surgery and underwent postoperative lower limb deep vein color Doppler ultrasound examinations. Among them, 40 patients experienced lower limb deep vein thrombosis (VTE) following surgery. The analysis encompassed various factors, including patients' demographic information, existing medical conditions, clinical details pertaining to the surgery, and results of laboratory tests. Specifically, these aspects included: ① Essential admission particulars, such as age, gender. ② Pre-existing medical conditions such as hypertension, diabetes. ③ Clinical data associated with the surgery, including the specific type and location of the fracture, cause of injury, duration between injury and admission, surgical techniques employed (such as acupuncture injection, massage therapy, acupuncture and moxibustion, traditional Chinese medicine treatments, etc.), method of anesthesia, and duration of surgery. ④ Laboratory results, covering complete blood count (CBC), coagulation function, and C-reactive protein (CRP). CBC parameters comprised white blood cell (WBC) count, platelet (PLT) count, among others.

### 4.8.2 Main Results

By conducting 10-fold cross-validation, we obtained the validation results for the model. Developed within the HPExplainPro framework and utilizing the CatBoost algorithm, the prognostic prediction model named "Zhongjing" achieved an impressive AUC of 99.30% on the validation set. This outperformed both XGBoost (AUC=90.00%) and the transformer deep learning model (AUC=89.90%). The optimal parameters for the transformer were determined as follows: d_model=128,

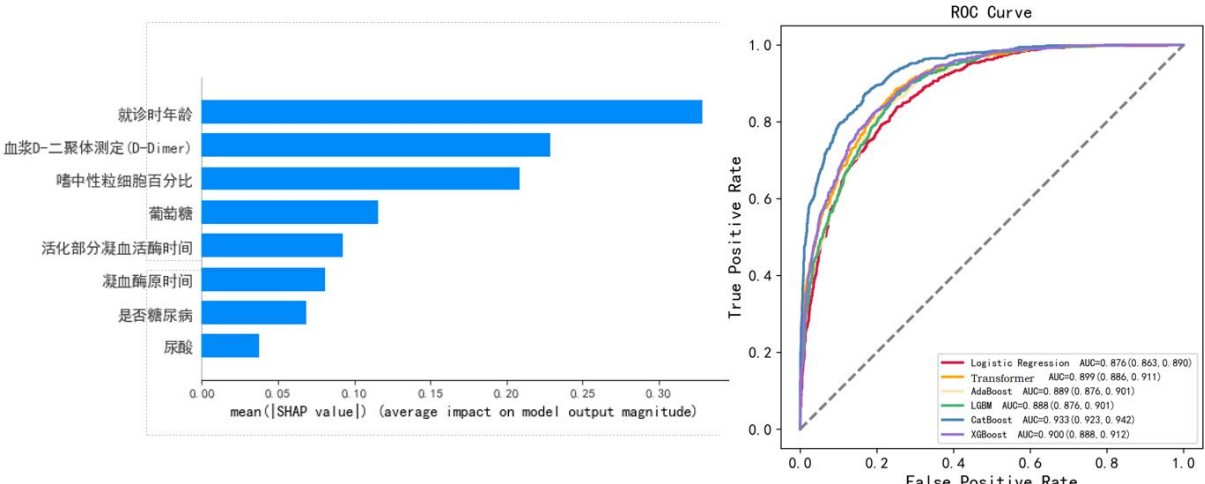

**Figure 8: Prognostic Prediction Model "Zhongjing" for Traditional Chinese and Western Medicine in Arthritis.**

nhead=4, num_encoder_layers=2, num_decoder_layers=2. Incorporating insights from prior experiments conducted by our team, we observed that when dealing with small sample data in the training set, the performance of the transformer model fell behind that of traditional machine learning models. As illustrated in Figure 8, the AUC curve of our model underscores the robust generalization capabilities of HPExplainPro. Additionally, our study identified several risk factors for postoperative lower limb deep vein thrombosis (DVT) in osteoarthritis patients, including 'plasma D-dimer measurement (D-Dimer)', 'glucose', 'prothrombin time', 'activated partial thromboplastin time', 'uric acid', 'diabetes status', and 'age at diagnosis'.

### 4.8.3 Potential biomarkers for arthritis prognosis in both Chinese & Western medicine discovered by DeepXplain

Various factors were observed from Figure 8, including age at diagnosis, plasma D-dimer measurement (D-Dimer), percentage of neutrophils, glucose levels, activated partial thromboplastin time, prothrombin time, diabetes status, and uric acid levels. Age plays a pivotal role in determining the effectiveness of osteoarthritis treatment. As individuals age, cartilage degeneration increases, and joint function declines, potentially resulting in less effective treatment compared to younger individuals. Younger patients may demonstrate better self-repair capabilities and treatment response. D-Dimer, a breakdown product of fibrin, typically increases in association with thrombosis or a hypercoagulable state. In osteoarthritis, D-Dimer levels may correlate with the severity of joint inflammation and blood circulation status, affecting treatment outcomes. Neutrophils, primary inflammatory cells, often indicate inflammation when their percentage increases. Higher neutrophil percentages in osteoarthritis patients may suggest more severe joint inflammation requiring stronger anti-inflammatory treatment. Blood sugar levels can affect the nutritional supply and repair capacity of joint cartilage. Elevated blood sugar levels may exacerbate joint cartilage damage and influence treatment efficacy. Activated

partial thromboplastin time and prothrombin time reflect the status of coagulation function. Abnormalities in coagulation function may lead to intra-articular bleeding or thrombosis, impacting joint function and treatment outcomes. Patients with diabetes commonly experience vascular changes and microcirculation disorders, worsening osteoarthritis symptoms and affecting treatment outcomes. Diabetes may also modify drug metabolism and distribution, affecting treatment efficacy. While hyperuricemia may be associated with gouty arthritis, its role in osteoarthritis is unclear. Some studies suggest that high uric acid levels may exacerbate joint inflammation and cartilage damage, affecting treatment outcomes. However, the surgical approach in traditional Chinese medicine, including acupuncture injection, massage technique, acupuncture and moxibustion, and Chinese medicine treatment, does not significantly impact prognosis.

In summary, factors influencing the prognosis of osteoarthritis treated with traditional Chinese and Western medicine include age at diagnosis, plasma D-dimer, neutrophil percentage, blood sugar levels, coagulation function, diabetes, and uric acid levels. During treatment, comprehensive consideration of these factors and the development of personalized treatment plans are essential to improve treatment efficacy and prognosis.

## 4.9 Clinical Application of HPExplainPro

The study described earlier has witnessed the integration of COVID-GRAM into numerous hospitals throughout Hubei Province, including Huoshenshan Hospital. This integration has solidified its reputation as an accurate and practical clinical tool widely embraced by the medical community. Inspired by the success of this endeavor, our research team collaborated with Hunan Provincial Tumor Hospital to apply the HPExplainPro model in the realm of breast oncology. Upon testing with a dataset comprising 100 cases, the model demonstrated an accuracy rate exceeding 90%. Among these cases, 88% represented breast

cancer patients with a 5-year survival rate, while the remaining 12% comprised patients who succumbed to the disease [20].

The effectiveness of the 5-year survival prediction model for breast cancer in clinical settings may not meet expectations due to various factors [21-23]. Firstly, breast cancer patients exhibit significant individual variations [24], including age, gender, physical health, lifestyle choices, comorbidities, and other factors [25]. While these factors can influence patient survival rates and treatment outcomes, they may not be fully accounted for by the model. Secondly, although the model incorporates features such as marital status, parity, lymph node metastasis, HER2 [26], Ki67 [27], staging, surgical history, and recurrence status, it may still not cover all factors affecting breast cancer's 5-year survival comprehensively. Genetic variations [28,29], other biological markers [30], environmental influences[31-33], and patient lifestyle habits [34] can also significantly affect survival rates. Clinical data may face quality issues like missing, erroneous, or inconsistent data, impacting the model's training effectiveness and prediction accuracy. For instance, some breast cancer patients may be lost to follow-up, resulting in incomplete data on recurrence status. Complex models may lead to overfitting, where they perform well on training data but poorly on new data.

To tackle these challenges, future efforts will concentrate on identifying additional factors related to breast cancer survival through further research and clinical observation, integrating them into the model.

This study's main contribution lies in uncovering multiple potential biomarkers linked to the prognosis of various diseases using the advanced tool HPExplainPro. These findings offer fresh insights into early disease detection, treatment strategy development, and prognosis evaluation, charting a course for future medical investigations. NETs and NLR, two insufficiently validated prognostic markers for lung cancer, have been successfully pinpointed. As indicators of immune response, abnormal levels of NETs and NLR may closely correlate with lung cancer progression and metastasis, aiding in more precise prognosis assessment and treatment planning for lung cancer patients [35,36]. Biomarkers NLR and BNP associated with critical COVID-19 illness have been identified, holding significant significance for promptly identifying high-risk patients, optimizing treatment approaches, and reducing mortality rates [37,38]. Biomarkers plasma D-dimer and uric acid levels [39,40] post combined traditional Chinese and Western medicine treatment have been discovered, providing scientific backing for refining treatment strategies in the future. Through HPExplainPro, this study has effectively identified numerous potential biomarkers linked to the prognosis of diverse diseases, offering substantial scientific value and fresh guidance for clinical practice. With further exploration into these biomarkers' mechanisms, more breakthroughs are anticipated in preventing, diagnosing, and treating related ailments.

## 5 CONCLUSION

We have designed and developed a sophisticated deep interpretable learning approach tailored for pan-cancer prognosis prediction, dubbed HPExplainPro. This framework brilliantly fuses transformer technology with interpretability methods, enabling it to deliver pinpoint-accurate disease prognosis predictions. Through meticulous experimentation, we have delved into clinical data pertaining to three cancers with elevated incidence rates: lung cancer, breast cancer, and liver cancer. The findings garnered from our investigations unequivocally reveal that HPExplainPro surpasses other cutting-edge methodologies in prognosis prediction. Remarkably, the versatility of HPExplainPro transcends solely cancer prognosis. It holds vast potential in prognostic predictions for COVID-19, analysis of traditional Chinese medicine treatments, and numerous other diseases. Leveraging the power of Deepxplain analysis, we have successfully dissected the model's feature utilization, presenting personalized and visually intuitive results. This innovative approach effectively dismantles the opaque nature of machine learning outcomes, offering biomarker insights for cancer prognosis that serve as invaluable references for clinical auxiliary diagnosis.

Throughout the entire research trajectory, from the initial conception of the method to its experimental validation and clinical implementation, HPExplainPro has exhibited remarkable coherence and synergy. During the experimental verification phase, the HPExplainPro framework exhibited exceptional performance across diverse cancer datasets, further underscoring its robust generalization capabilities through cross-disease validations. Ultimately, the clinical deployment of this innovative methodology not only validates its practical efficacy but also illustrates the seamless transition from theoretical conjecture to tangible real-world application.

## ACKNOWLEDGMENTS
This work was supported by the National Natural Science Foundation of China (Grant No. 61773157, No. 82274543), the Natural Sciences Foundation of Hunan Province (Grant No. 2021JJ30139, No. 2023JJ30471).

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
