# OpenReview forum: "HPExplainPro: A Framework for Pan-cancer Prognosis Prediction Based on Deep Interpretable Learning"
_KDD.org/2024/Workshop/AIDSH — KDD-AIDSH 2024 Poster_

### Official Review · Reviewer_Zef7 · 2024-06-12

**Rating:** 5
**Confidence:** 4

**Review:**

The authors introduce HPExplainPro, a deep explainable learning framework designed for pan-cancer prognosis prediction.
The framework integrates a deep learning model based on expert knowledge, a data-driven feature fusion approach, a triple feature selection technique, a heterogeneous classifier, and a secondary learning probability error integration model. A key component of HPExplainPro is the Deepxplain module, which uses both global (DeepSHAP) and local (LIME) interpretation methods to provide insights into the model's decision-making process. The framework's performance is demonstrated through the construction of predictive models for lung cancer immunotherapy response, breast cancer 5-year survival, and liver cancer local progression outcomes using datasets from Chinese hospitals.

However, here are some suggestions:
1. The novelty is limited. It seems that SHAP and LIME are both existing methods instead of the innovative designs. This may influence the technical contribution of this paper.
2. When we talk about the interpretability of neural network, we often ask for the explanation of the model parameters. It seems that there is no such "interpretability", but just explain the causality of each input features. So what about using some other interpretable models like AdaCare [1]? Attention mechanism is a good way to reach your goal.
3. The compared baselines are relatively insufficient. Please include more SOTA baselines such as AdaCare, ConCare [2], SAFARI [3], etc. for comparison.
4. Writing: The typesetting on Page 4 might be incorrect. Please check.

---

### Official Review · Reviewer_48hM · 2024-06-17
**HPExplainPro: A Framework for Pan-cancer Prognosis Prediction Based on Deep Interpretable Learning**

**Rating:** 3
**Confidence:** 4

**Review:**

**Summary**
The paper designed a deep interpretable learning framework for pan-cancer prognosis prediction, named HPExplainPRo. The frame work integrates feature selection, prediction model development and mainly focuses on the model explainability.

**Strength and weakness**
**Strengths**
- The models achieve great performance.
- Combine explainability method to the deep learning methods.

**Weakness**
- Writing should be improved and formatting should be adjusted.
- The Chinese in the summary plot of Figure 8 should be changed to English or annotated with English.
- The implement of cross-validation method seems to be problematic. The paper states a 10-fold cross-validation method. However, the paper splits the whole dataset into 10 parts and further uses a K-fold cross-validation for each iteration. This seems not how cross-validation works. 10-fold cross-validation should split the dataset into 10 parts and do the training and testing for each iteration. Cross-validation estimation is an estimation of true prediction error and a way to tune the hyper parameters of the models.
- About the external validation in section 4.8, it seems the paper also conducts 10-fold cross-validation on the dataset. This operation seems not to be an external validation which means to apply established model to a new dataset with a  same or similar distribution to the dataset used for model training.

**Questions**
- In the second paragraph of section 2 (Related work), it should be a citation for what Huang Yan said.
- For handling missing values, the paper suggests to exclude samples with over 50% missing values, any reference for the 50% criterion?
- The proposed method suggests to use cnn + transformers, however, it does not explain how these two methods are applied on the datasets and how do both methods combine together.
- In Table 4, the prognostic prediction results seem to be the results of all the samples in the datasets rather than test dataset. Further, since the paper apply 10-cross validation, there should be confidence interval when report the results.

---

### Decision · Program_Chairs · 2024-06-28

Accept (Poster)